# When Digital Doesn't Work: Experiences of Co-Designing an Indigenous Community Museum

Meghan Kelly [1,*] and Simone Taffe [2]

1 Faculty of Arts and Education, Deakin University, Burwood 3125, Australia
2 School of Design and Architecture, Swinburne University of Technology, Melbourne 3122, Australia; staffe@swin.edu.au
* Correspondence: meghan.kelly@deakin.edu.au

**Abstract:** The challenges to implement digital technologies in community-based projects are exposed in a case study co-designing an indigenous Community Museum, situated in the Kelabit Highlands of Borneo, Malaysia. Over a five-year period, this co-design project consisted of field trips, community engagements, and creating a documentary film and an inaugural exhibition in the newly constructed Kelabit Museum. This article highlights the limitations of digital technologies in museum contexts. Co-designing with stakeholders resulted in the decision to take a non-digital approach to the museum development to encourage greater community agency and prevent disengagement, as it incorporated heritage values in local community developments and cultural tourism plans. The findings demonstrate that community self-determination conflicted with preconceived outcomes, resulting in a need to re-evaluate the goals of the project. Instead, the ambition of cultural heritage preservation that maintained community participation emerged as the central goal. Removing the focus on a digital solution expanded community participation, which is a finding that should be used to frame other community cultural developments.

**Keywords:** co-design; community agency; digital technology; museum

## 1. Introduction

This article reflects on the expanding support for digitally enhanced museum experiences and demonstrates how, regardless of the increased adoption of digital technology to shape the user experiences of visitors, digital does not always work. This article presents the case study example of the Kelabit Community Museum project situated in the Kelabit Highlands of Borneo, Malaysia. In this study, digital technology was abandoned due to the location of the museum and the capacity of the community who had digital interest, but chose to focus on other priorities due to local constraints and limited access to essential resources. The authors argue that, with increasing digital currency, we must not lose sight of how to achieve an enriched user experience when contemporary digital practices are not available to curators and cultural caretakers.

As an overview of the project, the Kelabit Community Museum project was initiated by the Kelabit community and their strong need to preserve their cultural heritage. The project developed from an understanding of how the Kelabit community's traditional tangible and intangible knowledge was being lost in the transformation of the Kelabit community by progress. A specific aim of the Kelabit community was to use the museum as a means of harnessing heritage resources that were appropriated by others in the colonial past. Extensive details of the project have been published in a book titled *Museum Development and Cultural Representation: Developing the Kelabit Highlands Community Museum*, co-authored by Jonathan Sweet and Meghan Kelly. As outlined in this book, the Kelabit community predominately lives in the small, isolated town of Bario in the highlands of Borneo and a larger city, Miri, located on the Baram River in Sarawak's northeast. The Kelabit community

is also dispersed, residing in Kuching, the capital city of Sarawak, and Kuala Lumpur, the capital of Malaysia. Kelabit are designated as an indigenous tribe, in part because they have a distinct language that distinguishes them from other native tribes.

Culture in Malaysia is linked to centralised government conservation policies and processes that have often neglected ethnic, religious, and indigenous communities. The local, spiritual complexity of historical sites in the Kelabit Highlands did not fit the broader constructs of national identity as defined by the Malaysian government. Although their ethnicity and indigeneity are recognised in Malaysian law, the group have little agency in matters of government decisions as they are a Christian and indigenous minority. For this reason, the community chose to organise themselves and take ownership of preserving their unique cultural heritage and advocate for their cultural interests. Although one of the smallest indigenous groups in the Sarawak region, numbering approximately 6000 people, the Kelabit community has become renowned for its economic and professional success and has come to understand how marketing, tourism, and education can contribute to the community's own development goals [1] (p. 5). The community recognised that the Kelabit Community Museum project would provide the opportunity for the Kelabit community to assert its authority over the representation and commodification of heritage, "while also creating environs inclusive of new and diverse voices of expertise and authority" [2] (p. 96).

The Kelabit Community Museum was considered an additional component that fit with the desire to develop the town centre as a commercial and cultural precinct, and build upon previous cultural development work that had occurred in the town [3]. It was anticipated that this development would enhance the existing and neighbouring complex used for congregation and recreation, including the town hall, local shops, and cafes, and be used to support the market for locally made traditional craft objects and souvenirs. The Kelabit Community Museum was named *Teripun*, meaning "safe space" in the Kelabit language, with the construction of the built environment completed in 2016 [4].

The development was a five-year co-design project to reframe intangible cultural knowledge into something tangible. The focus was on the transformation of ideas and opinions of a predominantly verbal cultural group into physical designed artefacts. The project stemmed from an extensive list of aims and ambitions that were articulated in a discussion document created by Nikki Lugun in 2011 on behalf of the Rurum Kelabit Sarawak [5]. The Rurum Kelabit Sarawak (RKS) is a non-political entity that operates and conducts its business as the Society of Kelabits in Sarawak under the Societies Act 1966 [4]. The RKS is distinct from the official system of local governance. It is a self-funded, non-governmental, cultural organisation that provides a mechanism through which the community can advocate for their cultural interests. In her document, Lugun included a priority list of activities aimed at preserving the Kelabit culture: educating others and passing on traditions; preserving materials in personal collections; documenting culture and indigenous knowledge; assisting tourism and commerce ambitions; and transferring technology skills. To achieve this, the Kelabit Community Museum project aimed to utilise the intellectual capital that resided within the community and determine the most appropriate means to document and communicate Kelabit cultural heritage.

The discussion document created by Lugun adeptly outlined a vision that was centred on reclaiming cultural heritage assets for the benefit of the Kelabit community. It also identified some gaps in the professional expertise of the community that would need to be addressed to achieve these goals.

The community members realised that they did not have the resources to achieve the goals of the Community Museum project alone. They would have to approach various organisations to assist them, recognising that many of these organisations could bring significant intellectual, archival, and technological resources to such a project [5].

The Kelabit community invited a multi-disciplined research team from Deakin University, including student representatives, to incorporate contemporary museum and co-design theory to drive the development and establish the framework for a sustainable and engaging museum experience. The Deakin University research team was tasked with facilitating

the co-design process, and it was acknowledged from the beginning that the final decisions would rest with the Kelabit community to ensure they were community-led.

Working with the community over the five-year period using a co-design methodology, it was revealed that although the use of digital technology was a primary motivator for capturing a self-determined representation of Kelabit culture, realising and maintaining that vision was not achievable. The end result saw the community move away from technologically driven ideas, as it resolved to not have technology overshadow the core ambition of cultural heritage preservation that was self-determined and maintained community participation.

## 2. Background

### 2.1. Digital Technology in Museums

Museums, as flexible, growing, and organic entities, are closely connected to the culture, economy, and demographics of the communities they serve [6]. They are an institute intent on educating the public through social practices where people engage with their environment and each other. Visitors are invited to join the collective interest in culturally specific tangible and intangible knowledge shaped by the community and personal contexts [7].

Museums are understood to be "invaluable resources because of their potential to foster local identity in a time of increasing globalisation and to be representational bodies engendering a sense of belonging to the groups who make and use them" [8] (p. 81). In our case, the Kelabit community and its governing body, the Rurum Kelabit Sarawak, were well-versed in the advantages of a self-determined commodification of their cultural heritage. They understood their cultural heritage would serve as an important contributor to ensuring a sustainable culture, maintaining and strengthening community connections and creating a healthy, strong community [9] (p. 2). The museum would be used to assist in the development of regional renewal, empower community members through a reflection and representation of history and realities, and provide a commentary on how these realities have changed and evolved.

Previous studies show that designing museums and exhibition settings is complex. Discussions include the advantages and disadvantages of interactive technology in museum design [10] and the role of social media usage in museums [11]. Supporting strategic initiatives is deemed a way to attract new audiences [12], and digital technologies are increasingly present in museum exhibitions as both a communication tool and as an object itself [13] (p. 27). Digital technologies are a major focus in museum design strategies, with mobile devices championed as providing a new context for learning and opportunities for users to engage with content [14] (p. 51). As a result of their implementation, "the learning landscape is dramatically widened beyond stable learning objects" [15] (p. 225). When museums adopt particular technologies, especially mobile tools and applications, they begin to create a multi-sensorial environment in which the visitors are informed during their visit without the need to stand still. Digital devices are the connective device to relational heritage information, supporting the visitor experience and helping to develop visitor comprehension [16] (p. 163). They cater to the active learner, presenting a variety of ways in which the visitor can explore and learn in a personalised and contextualised way [17] (p. 842). In general, the shift from pure content delivery to a process of social interpretation (with an emphasis on user-centred design processes) has resulted in an upheaval in communication strategies within museums [18] (p. 805).

Noting the initial introductory document of Lugun stating that the use of technology was an aspiration for the Kelabit Community Museum project, aiming to include digital technology is an understandable objective when you consider that cultural centres compete with other venues for visitors' recreational time [19]. Roussou and Katifori argue that using technology in a museum setting comes with a number of challenges that need to be considered during the design and/or deployment stages of development. Their research identified, for instance, that visitors do not like staying in one place for long and enjoy

the use of dynamic elements enriched with visual, audio, or motion effects. Roussou and Katifori also noted that the creators of content need to be careful that the digital devices do not overpower the overall storytelling experience. Yet, the focus on digital technologies to improve the visitor experience neglects to include how the use of these technologies can be employed by smaller community-based museums who compete with larger, well-established institutions, leading to a loss of agency in smaller community-based museums.

### 2.2. Co-Designing Museums with End-Users

Co-design is not just designing by, for, and with potential users [20], but, more broadly, also collaborating across different stakeholder groups and external parties [21]. Along with the development of digital technologies, there has been a concurrent increase in co-design and collaborative processes to engage stakeholders in the design process. Co-design introduces to designers and stakeholders new challenges such as negotiations across community members with differences in languages, values, and concerns [22], and different levels of digital understanding and expertise [23,24]. Co-design is a design process enriched by many perspectives and understanding collective objectives, which may, in turn, result in the framework that drives the project being the framework that requires significant reconsideration [21]. In this case study, the researchers and community members were open to consider how the museum would be understood by the community, recognising that the community members would ultimately determine the outcomes.

Therefore, the role of the designer translates into that of a facilitator, where designers use their knowledge to help end-users fulfil their needs and to empower them in the design process [25–28]. A range of terms have been used to describe the designer's role of facilitation. Some talk of "bridge-building" between the worlds of end-users and designers to create something new from the combination of designers' technological knowledge and end-users' local tacit knowledge [25,29]. For Wai and Sui, the designer no longer aims to deliver fixed solutions, but rather facilitates conversation with end-users [30]. Wai and Sui argue that the role of facilitator "allow[s] more flexibility for users to actualize designs and participate in the decision process". Luck [26] depicts participatory design as a social process that transfers end-user knowledge to the designer, who then integrates this knowledge to the design process for the end-user's benefit. For Spinuzzi [28], the design process becomes a forum for negotiating different design options. Friedman [31] depicts the designer as a "synthesis" who solves problems by understanding the range of talents required to address them. For Frascara [32], the role of the designer could be seen as a guide, or coordinator, supporting end-users and decision makers to achieve creativity. For this case study, the role of the research team was to facilitate design ideas to promote discussion so that the community could see how their ideas may unfold. The tangible outcomes of the community workshops were used to inform further decision making.

## 3. Kelabit Community Museum Case Study

### 3.1. Using the Co-Design Method

Our co-design process involved working *with* the Kelabit community and the Rurum Kelabit Sarawak. The Kelabit Community Museum project necessitated a research method that responded to the complexities of the museum design process. Knowledge was constructed through reflecting on each situation as it arose, allowing this process to inform the next stages of the project. Ten visits were made to Malaysia, including three field trips and three formal visits with four informal consultations undertaken when academics were travelling abroad for other research projects. An overview of the engagements can be seen in Table 1, below. The techniques of notetaking and photography were used to record the workshops' proceedings. During the co-design workshops, Deakin University researchers and students acted as notetakers recording general observations, photographing the activities and design outcomes. The data sources included: photographs, meeting notes, drawings, personal conversations, notetakers' notes of three co-design field trips, email correspondence between all stakeholders, and journal entries documenting the community

forums and meetings. Everything created was provided to the community to assist with their decision making. For analysis, all of the data sources were organised chronologically and thematically coded. Extensive details of the engagements can be accessed in the book publication of the project [33].

**Table 1.** Field trips to the Kelabit Highlands over five years.

| Field Trips | Goal |
|---|---|
| 2011 Visit | Introductions |
| 2012 Field trip 1: 15–28 June | Community-wide scoping exercise |
| 2013 Visit | Review and planning |
| 2014 Field trip 2: 15–30 January | Architecture and built environment field trip |
| 2015 Visit | Review and planning |
| 2015 Field trip 3: 3–15 December | Branding and inaugural exhibition field trip |
| 2012–2015connor xu4 in-between visits | Brief consultation visits were made during the project. These occurred as academics were travelling abroad for other research projects. |

### 3.2. Three Co-Design Field Trips

Initially, the project was developed as a result of discussions between the Kelabit community and Ms Jan Drew, an independent educational consultant based in Malaysia. Representatives of Rurum Kelabit Sarawak, the governing body of the Kelabit community, through Jan Drew, contacted Dr Jonathan Sweet and invited him to visit the region to discuss how a Community Museum may be achieved. The project related to Jonathan Sweet's cultural heritage research and development interests in the region. He has studied the history and practices of museology in Sarawak and has also been actively involved in UNESCO and ICCROM projects in other parts of southeast Asia. This led to an invitation to conduct a scoping field trip in June 2012, with Dr Sweet leading a small team of Deakin University students who visited the region to assess the feasibility of establishing a Community Museum in Bario [34]. It was essential to determine the level of community support for the development and the process by which the data collection and documentation of the community's interests and cultural assets could be managed. A co-design process was designed where evidence was gathered from three field trips to the highlands in the Borneo region and a series of comprehensive community consultations in Bario, Miri, and Kuching.

### 3.2.1. Co-Design Field Trip 1

In the first field trip, 15–18 June 2012, we began with an ethnographic (direct observation) and interview process, which formed the basis of the information gathering where between 15 and 30 Kelabit community representatives attended each session. The Kelabit community included those who did not live in the highlands, but returned to maintain their personal community and historical connections to the region and those who were based in Bario, the central township in the Kelabit Highlands.

In June 2013, a team of academics consulted with the broader Kelabit communities of Miri and Kuching to refine and develop the aims and ambitions of the project [35]. The participants in the community discussions used whiteboards and large sheets of paper to structure their vision. Documented were specific questions concerning aspects of interpretation and community engagement, policy and governance, the availability of resources, and infrastructure. One of the main worries that emerged was understanding ways of managing the use of privately owned artifacts and issues of human resources. Other questions included who would maintain the programmes, what kinds of employment opportunities might be available, and how the museum would work in circumstances where the availability of electricity was inconsistent.

### 3.2.2. Co-Design Field Trip 2

The success of the first field trip led to two subsequent field trips and co-design engagements. The second field trip, 15–30 January 2014, was with a multidisciplinary team of academics and architecture and design students who co-created concept designs for the built environment with the community. The information gathered and the designs created demonstrated to the wider community how the museum may work. The community used these designs to consult more broadly with the community and with local architects to continue to develop their vision. Strong attendance at the community consultations during these co-design workshops indicated a desire to investigate options for displaying historical assets in both digital and non-digital formats. Discussions included the desire for an online presence, digital and non-digital methods of engagement with visitors to the museum, and ways people may be able to interact with artifacts. The community held a strong interest in creating interactive exhibits supported by digital technologies with touch, feel, and in-depth content, presenting a range of narratives.

### 3.2.3. Co-Design Field Trip 3

Drawing on the previous field trip outcomes, the final field trip, 3–15 December 2015, led to the research team being tasked with creating a comprehensive display of Kelabit identity for an inaugural exhibition. The exhibition displayed in *Teripun* promoted further discussion and debate about content creation and presentation.

## 4. Findings

### 4.1. Benefits of a Digital Solution in the Highlands

The co-design process translated the narratives drawn from the community participants into tangible, visual expressions that were presented back to the community. The outcomes were used to promote an ongoing dialogue on ways in which to achieve authentic Kelabit representation. The interpretive material produced was used to question and re-evaluate a meaningful exemplification of Kelabit identity. Concerns over a lack of agency in representation were alleviated when, for the first time, the village had a Kelabit-generated view of their history presented back to them that could be evaluated. This view was important to help the community have a collective conversation about how they understood their history. The outcomes helped to differentiate between fact and fiction, and as stated by one attendee, it served as a powerful form of community consultation itself. As a result, the co-design process allowed the community to capture the content in a display that was able to serve as a starting point for further debate and discussion.

The large mass of information and opinions captured during the co-design workshops led the research group to acknowledge that digital outcomes would greatly benefit the Kelabit Community Museum project. Three particular themes emerged where digital solutions would offer great advantage: the first was to capture multiple narratives, the second was to allow for multiple languages to be presented, and the third was to address the concerns regarding the use of personal collections in the museum.

To explain this further, tensions within the community emerged during recollections of details, as different people held different views of the same events and stories. There was great deliberation between what was deemed fact and what was fiction. At times, this process was quite confronting for the participants, especially when long-held understandings or orthodoxies were challenged by other participants. The overwhelming importance to document these understandings was clearly evident in each of the community consultations; however, it was difficult to document the many different interpretations of a single story in a print-based format. Initially, the research team believed in the benefits of incorporating digital technology into the museum design. Acknowledging digital technology allows for the presentation of multiple stories, personal histories, and cultural contexts, allowing the explanation of why an object is important to people in different ways [36].

In addition, it was requested that the text be made available in the English, Malay, and Kelabit languages. Although translation into Malay was also a priority, writing content

in a third language was not possible due to the availability of a translator and the limited space of a print-based exhibition. Adding to the complexity, the written language of the Kelabit community had only recently been captured and was undergoing a process of re-evaluation. An issue arose as one segment of the community did not want to change the traditional Kelabit written structures, and another was in favour of creating a written form more closely reflecting the spoken. Unaware of these tensions, the research team worked with a translator who used the new, contemporary approach to the written Kelabit language, much to the criticism of some who attended the inaugural exhibition. If there were opportunities to present the information digitally, the option for the end-user to select from multiple languages—including the understandings of the traditionalist and the modernisers within the community—could be provided.

Lastly, one of the strongest concerns identified, and is still ongoing, was the threat the museum may bring to personal and family collections and the ownership of assets. The concept of borrowing content and documenting, presenting, and returning precious items brought wariness to the community. As identified in the initial planning documents, the digital strategies of recording, documenting, and presenting precious family artefacts would serve as a solution to concerns over ownership and the protection of personal assets.

### 4.2. Challenges of Adopting Digital Technology

Although the Rurum Kelabit Sarawak, Kelabit community, and Deakin University research team agreed on the benefits of working with digital technology, a number of challenges became evident. With these factors in mind, progressively digital factors were abandoned and, instead, students created a substantial, albeit temporary, print-based exhibition for the community. With the ambition of the community and Deakin University research team to maximise digital technologies in this exhibition, all digital equipment was taken to Borneo for use by the research team. This included computers, a scanner, a colour printer, digital cameras, and a laminator to protect any printed collateral from moisture. There was no equipment available for use in the remote location unless it was brought in by the research team. Handheld devices were not included in the package of equipment taken to the region due to the luggage limitations and weight restrictions flying from Miri to Bario.

Working with the technology on location exposed some significant challenges, in particular the capacity of the solar power infrastructure in the region and the lack of strength to charge the computers and power the printer. Achieving the design outcomes required multiple appliances running consecutively. A petrol generator was necessary, leading to an increased demand for petrol, and several petrol purchases made using mobile containers were required. Power loss was common, and content not saved regularly was lost. Bikes were hired by the students to conduct research and community visits, yet punctures were also common. The exhibition took place during the rainy season, so weather, transportation, and accessibility impacted the outcomes. The digital appliances needed to be protected from the extensive rain and humidity, and the students found that local bugs were attracted to the engines of the computers. The students explored the use of recording devices and the presentation of video content; however, community members found the technology intimidating and felt confronted by its use, impacting on the recording of content.

The scanner, printer, and laminator were donated to the museum for ongoing use. The research team provided lessons and support information to the local community members based in Bario, but, anecdotally, once our role in the project was complete, the research team became aware the community members were able to work productively with the equipment again. This highlights how the limited capacity of the community was revealed through the co-design workshops, noting many local members of the community were unfamiliar with how to manage and maintain digital content. The community members were excited to work with the students to develop the content, but were apprehensive

about engaging directly with the technology themselves, preferring to use the students as a conduit to achieve outcomes.

The exhibition design created by the Deakin University research team did not directly draw on the resources of the community, and instead used images in the narratives to catalogue and present the content. This proved a comfortable option for the community, but was limited in its ability to fully recognise and capture the diversity of the available artefacts.

## 5. Conclusions and Future Directions

The Kelabit Community Museum project was greatly enriched by the many stakeholder perspectives that emerged in the three co-design field trips. Driving this project was the collective objective to capture and preserve Kelabit cultural heritage, and it was agreed that a priority for the success in achieving this goal was the participation and engagement of the Kelabit community who were located in Bario and those who lived outside of the highlands. The framework guiding the discussion was also the framework that can be seen to constrain the outcomes, and resulted in a shift in thinking away from using digital technologies as an exhibition solution [21]. Despite the understanding of the benefits of using digital technologies, the lack of digital competencies, resources, and the unique environment of the community members based in the highlands of Borneo meant that the focus on technologies and their capabilities would derail the objectives of the community.

This understanding only emerged slowly, and over time, through the co-design engagements. With this realisation, a shift also occurred within the narrative of the Rurum Kelabit Sarawak and senior community members from the concept of a museum to that of a cultural centre. Although this change in attitude evolved over time, the outcome was a significant amendment from the primary concern of heritage conservation to benefit the community and attract tourists, to a more major concern of bringing the community together to use the space for events such as talks and art exhibitions, and as a space to gather knowledge and host educational programmes. The adjustment came from a realisation that some of the ambitions and requirements of cultural preservation would not be achievable without digital technology, which was limited by the local constraints. The initial requirements of determining museum governance and curatorial responsibilities, including addressing concerns over the use of personal and family collections and the ownership of assets, were all discussed with digital technologies in the forefront of thinking. Over time, it was identified that these discussions had transformed into encouraging engagement and foregrounding use by the community, with the limitations of using digital technology forcing the reconsideration of this objective.

The repatriation of acquired artefacts is a concern for many indigenous communities, yet an infrastructure to support the acquisition of and to suitably house such objects is difficult to achieve in a remote context, particularly where the climate impacts on preservation. In addition, the borrowing of personal objects for display and concern over the handling and protection of the artefacts could be mitigated using digital solutions. The introduction of more advanced technologies, currently identified as a challenge in the remote region of Bario, would add considerably to the development and preservation of Kelabit cultural knowledge and practices. It was resolved that future acquisitions and preservation practices would need to be considered by the community as technology advances.

In the future, with support from external participants, the aim for the Kelabit Community Museum is to build a permanent collection with secured exhibit cases and free-standing panels, supported by digital strategies. The digital recording of items, designed and documented with audience and visitor engagement in mind, could then be used to establish a circular process of moderation and evaluation that would support the Kelabit community's attempts to manage authenticity and continue to shape representation. However, at this stage, the appreciation of the Kelabit culture is developing through community engagement in the museum space with a celebration of traditional songs, dances, language, and folklore.

In the case of co-designing the Kelabit Community Museum, digital did not work. Working with the community over the five-year period using a co-design methodology



revealed that although the use of digital technology was a primary motivator for capturing a self-determined representation of Kelabit culture, realising and maintaining that vision was not achievable. The community moved away from the use of digital technologies, as it resolved to not have technology overshadow the significant ambition of cultural heritage preservation that would be self-determined through sustained community participation. Ongoing community engagement in the project was seen as a higher priority, which meant it became necessary to adjust preconceived ambitions as the project evolved. This led to a fundamental change in the overall goal of the museum from a focus on cultural heritage preservation to a focus on living community participation.

Looking broadly at what researchers can learn from our experiences, we offer two recommendations. Firstly, this research project highlighted the limitations of working with digital technology in remote regions where there are community, power, and resource limitations. We suggest that digital solutions are not a blanket approach worth pursuing in all cases, and recommend that design and research teams are open to abandoning digital technology if, during co-design workshops, it becomes clear that digital technology will limit sustained community engagement practices or be compromised by power and resource limitations. Community projects are only successful with ongoing community engagement and resource support, and without these, it became clear that the Kelabit Community Museum project would not be sustainable.

Secondly, the community dictated the outcomes of the project, even if some stakeholders could see a different direction, and this was an essential element in achieving success for the project. Having end-users determine the outcomes of a project is a fundamental component of human-centred design, but is also a difficult process for designers to fully embrace. With the Kelabit Community Museum project, the designers were required to relinquish their expert contribution to the project, and therefore some control, to allow the project to evolve and achieve outcomes.

We propose that in each community development project, there may be a sliding scale of appetite and feasibility for digital solutions. Co-design methodologies help to reveal tensions when pre-planned outcomes cannot be met and a readjustment of goals is required. Using a co-design process assists designers and communities to define and redefine the project goals and assess whether digital technologies are indeed the best solution.

**Author Contributions:** Conceptualization, M.K. and S.T.; methodology, M.K.; investigation, M.K.; resources, M.K.; data curation, M.K.; writing—original draft preparation, M.K.; writing—review and editing, S.T.; project administration, M.K.; funding acquisition, M.K. All authors have read and agreed to the published version of the manuscript.

**Funding:** This research was funded by the Australia–Malaysia Institute Program, grant number AMI-14/15-GG5 and the New Colombo Plan, Department of Foreign Affairs and Trade.

**Institutional Review Board Statement:** The study was conducted according to the guidelines of the Deakin University Human Ethics (HAE-13-114 Kelabit Highland Community Museum Development Project, 26 November 2013).

**Informed Consent Statement:** Informed consent was obtained from all subjects involved in the study.

**Data Availability Statement:** All data contained in the article.

**Acknowledgments:** We would like to acknowledge the generosity and contribution of the Rurum Kelabit Sarawak and the Kelabit community.

**Conflicts of Interest:** The authors declare no conflict of interest. The funders had no role in the design of the study; in the collection, analyses, or interpretation of data; in the writing of the manuscript, or in the decision to publish the results.

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
