# Peer review of "When Digital Doesn’t Work: Experiences of Co-Designing an Indigenous Community Museum"

_mti, doi:10.3390/mti6050034_

Round 1

Reviewer 1 Report

Among the reviewers from the first round, I was the one with the most positive view of this paper. So of course I still support this paper, and I do not  repeat my statements from the first round. 

Looking at the revised version, I can see that the authors have responded very constructively to the issues I have raised, so I am quite happy with the current state of the manuscript. Thanks to the authors for a quick and thorough revision.

Altogether, I am now supporting acceptance. I understand, however, that the other reviewers and the editors had further concerns, so it is top to them now to take a decision.

Author Response

Thank you to Reviewer 1 for their ongoing support of this paper. The editor's comments required two areas of clarification. Firstly, more detail on the Kelabit Community which has been added to pages 1 and 2 of the paper. Secondly, clarification of what researchers can learn from our experiences and the conclusion has been rewritten to offer a more transparent overview of lessons learnt. Please see page 9 for the revised text. We thank you again for your comprehensive review of this paper, noting each time it has offered significant improvement. Warm regards, Meghan (and Simone).

Reviewer 2 Report

I accept this paper in present form.

Author Response

Dear Reviewer 2.

Thank you for your support of this paper. We are pleased we were able to satisfactorily address your comments.

With regards

Meghan Kelly

This manuscript is a resubmission of an earlier submission. The following is a list of the peer review reports and author responses from that submission.

Round 1

Reviewer 1 Report

Although this article discusses an interesting topic related to co-design of indigenous community museums, the co-design project that has been undertaken seems to be largely a development project without much research implications.

The authors refer to adopting a co-design process for the project, and while they provide a literature review of co-design, in terms of their own project they do not describe their process very clearly. For instance, there is no information about how the co-design workshops were conducted,  what co-design methods were used, or how the process was recorded and analysed. The only information provided in terms of the process is how many meetings were held and the fact that some notes were taken along with photographs, etc. Without a detailed description of the co-design process and methods of data gathering and analysis, it is therefore impossible to evaluate the research novelty of the co-design process adopted and tested by the authors and how it could be used in other similar projects.

The other major shortcoming of this article is that while the authors/researchers have used some basic technology (e.g. computers, scanners, etc.) during their project to digitise artefacts, the final exhibits are produced as printed material. While there is nothing wrong with the use of such material in remote museums, this project does not address any research issues of relevance to the topic of this journal, which is about Multimodal Technologies and Interaction.

Reviewer 2 Report

This paper describes a co-design process to design an indigenous community museum in Malaysia. The paper "highlights the limitations of digital technologies in museum contexts."

Although I consider it interesting to discuss the limitations of digital technologies in museum contexts, I have serious doubts about the adequacy of this paper for this journal and about the level of detail about the descriptions of the limitations of digital technologies in this specific situation.

The paper does not discuss any limitation of specific multimodal technologies or interaction techniques and, as such, I would not see it within scope for this journal.

Considering, however, that it would be within scope, I think the paper does not contribute much to understanding the "limitations of digital technologies in museum contexts". There is a single section that addresses these (4.2 Challenges in Adopting Digital Technology), but, to me, this was confusingly written and I ended up not understanding exactly what were the challenges/limitations.

Some limitations seem to stem from transportation issues? "due to luggage limitations and weight restrictions flying from Miri to Bario"

Others seem to stem from local infrastructure. "Working with the technology in location exposed some significant challenges, in particular the capacity of the solar power infrastructure in the region and the lack of strength to charge the computers and power the printer."

(But then the account is mixed with challenges not related to digital technologies. "Bikes were hired by students to conduct research and community visits yet punctures were also common. The exhibition was during rainy season so weather, transportation and accessibility impact outcomes")

Other challenges seem to be attributed to the community's skills with technology, but then this sections states that the community was able to work productively: "once our role in the project was completed, the research team became aware the community members were able to work productively with the equipment again.

So I honestly did not understand the limitations of digital technology in this context. 

I think to understand those limitations, the paper should provide a technical description of the infrastructure available there, and more exact account of the reasons that led to the abandonment of digital technology.

Another concern is that the paper mentions a book (Museum Development and Cultural Representation: Developing the Kelabit Highlands Community 39 Museum) that has been published about this project, but the book is not provided for review. It is impossible to assess whether this paper adds anything significantly to what has already been published in the book.

Reviewer 3 Report

This is a paper which does not focus on technology at all. So, as a computer scientist, I was wondering whether I am competent to judge on the paper. However, since I have been involved in many projects bringing digital technologies to museums, I am very much interested in the topic. My personal experience is only from a different context, which is museums in technologically well-developed countries, so I found this report extremely interesting, and I think it will be important information for other colleagues as well.

The key sentence of the paper for me is the following one: "We recommend being open to abandoning digital technology if during co-design workshops it becomes clear that digital technology will limit sustained community engagement practices”. This is very much in line with the general approach of putting human aspects first, and technology aspects second, so I think this is an extremely helpful example for the relative (un-)importance of digital solutions. So, generally I like the paper very much and I am in favour of it being published.

There are just a few issues which I would like to be clarified for a final publication:

(1) The research was carried out up to 2015/16, which is about five years ago. Why is this work reported so late? I can imagine that the opportunity of the thematic special issue was an important factor, but still the readers in 2021 might be interested in what has happened in the meantime. I would suggest an update abut the current situation, also regarding the literature references.

(2) The paper is not fully clear about the main reason why digital technologies were abandoned. Technical limitations like unreliable power supply are well understandable. But I am wondering whether cultural issues like deciding "what was deemed as fact and what was fiction" were increased by the concept of using digital technology, or whether these are unrelated problems. Similarly, the mentioned issue that "borrowing content, documenting, presenting and returning precious items bought wariness to the community” at a first sight might be relieved by just using digital, and maybe even copied or abstracted, representations. I would like to see some more explanation on the relationship between technological and cultural issues here as well.

(3) I would love to see a critical limitation section where it is discussed which experiences may be transferred to other contexts, and which are specific to the concrete context. For instance, the fact that the small community led to an unstable basis for written texts in the community language will not apply to many other museum contexts.

There are also a small number of language problems (but not being a native speaker, I may also be misled here). Two examples are here:
- p. 5: "we began with an ethnographic (direct observation) and interview process formed the basis” - I think there is a "which" missing.
- p. 9: "in a remote locate” - do you mean "locale" or "location" instead?
Generally, the paper will benefit from a final language 
review.

Altogether, I see this as a very good contribution to the special issue entitled "Co-Design Within and Between Communities in Cultural Heritage". It fits almost perfectly with this title - I would have been more sceptical in the case of a general submission.